# Graph Piece: Efficiently Generating High-Quality Molecular Graphs with Substructures

## Abstract

Molecule generation, which requires generating valid molecules with desired properties, is a fundamental but challenging task. Recent years have witnessed the rapid development of atom-level auto-regressive models, which usually construct graphs following sequential actions of adding atom-level nodes and edges. However, these atom-level models ignore high-frequency substructures, which not only capture the regularities of atomic combination in molecules but are also often related to desired chemical properties, and therefore may be sub-optimal for generating high-quality molecules. In this paper, we propose a method to automatically discover such common substructures, which we call *graph pieces*, from given molecular graphs. We also present a graph piece variational autoencoder (GP-VAE) for generating molecular graphs based on graph pieces. Experiments show that our GP-VAE models not only achieve better performance than the state-of-the-art baseline for distribution-learning, property optimization, and constrained property optimization tasks but are also computationally efficient.

## 1 Introduction

Molecule generation is a task that aims to produce chemically valid molecules with optimized properties. It is important for a variety of applications, such as drug discovery and material science. Graph-based molecule generation models, which are robust to molecule substructures (You et al., 2018; Kwon et al., 2019), have gained increasing attention recently (Jin et al., 2018; Li et al., 2018a; You et al., 2018; Kwon et al., 2019; De Cao & Kipf, 2018; Shi et al., 2020; Jin et al., 2020b).

Graph-based molecule generation models typically decompose molecular graphs into sequential actions of generating atoms and bonds autoregressively (Li et al., 2018b; You et al., 2018; Li et al., 2018a; Jin et al., 2020b). While this decomposition is natural and straightforward, it inevitably ignores the existence of common substructures in molecular graphs, as illustrated in Figure 1. Compared with using atoms for generating molecules, using graph substructures for generating molecules have three potential benefits. First, using substructures can capture the regularities of atomic combination in molecules, and therefore is more capable of generating realistic molecules. Second, using substructures can better capture chemical properties, as there is a correlation between substructures and chemical properties (Murray & Rees, 2009; Jin et al., 2020b). Third, using substructures enables efficient training and inference. It is evident that using substructures to represent molecular graphs can result in much shorter sequences, therefore the training and inference process can be accelerated. As a result, we believe that models using substructures to represent molecular graphs can generate more realistic molecules with better-optimized properties efficiently.

Figure 1: Five high-frequency substructures in the standard ZINC250K dataset (Irwin et al., 2012; Kusner et al., 2017). The percentage means the ratio of molecules which have the substructure.

In this paper, we present an iterative algorithm to automatically discover common substructures in molecules, which we call *graph pieces*. Initially, graph pieces correspond to single atoms that appear in graphs of a given dataset. Then for each iteration, we count the occurrence of neighboring pieces in graphs and merge the most frequent neighboring pieces into a new graph piece. Since substructures can be seen as small molecules, we use SMILES (Weininger, 1988), a text-based representation for molecules, to efficiently judge whether two graph pieces are identical. To effectively utilize these substructures, we also propose a graph piece variational autoencoder (GP-VAE). Our model consists of a graph neural network (GNN, Scarselli et al., 2008) encoder and a two-step decoder. The two-step decoder first generates graph pieces auto-regressively and then predicts atom-level bonds between graph pieces in parallel. As a result, our GP-VAE decouples the alternated generation of nodes and edges, achieving significant computational efficiency for both training and generation.

We conduct extensive experiments on ZINC250K (Irwin et al., 2012) and QM9 (Blum & Reymond, 2009; Rupp et al., 2012) datasets. Results demonstrate that our GP-VAE models outperform state-of-the-art models on distribution-learning, property optimization, and constrained property optimization tasks, and are about six times faster than the fastest baseline.

## 2 RELATED WORK

**Molecule Generation** Based on different representations for molecules, molecule generation models can be divided into two categories: text-based and graph-based. Text-based models (Gómez-Bombarelli et al., 2018; Kusner et al., 2017; Bjerrum & Threlfall, 2017), which usually adopt the Simplified Molecular-Input Line-entry System (SMILES) (Weininger, 1988) representation, are simple and efficient methods for generating molecules. However, these models are not robust because a single perturbation in the text molecule representation can result in significant changes in molecule structure (You et al., 2018; Kwon et al., 2019). Graph-based models (De Cao & Kipf, 2018; Shi et al., 2020; Jin et al., 2020b), therefore, have gained increasing attention recently. Li et al. (2018b) proposed a generation model of graphs and demonstrated it performed better than text-based generation models on molecule generation. You et al. (2018) used reinforcement learning to fuse rewards of chemical validity and property scores into each step of generating a molecule. Popova et al. (2019) proposed an MRNN to autoregressively generate nodes and edges based on the generated graph. Shi et al. (2020) proposed a flow-based autoregressive model and use reinforcement learning for the goal-directed molecular graph generation. However, these models use atom-level graph representation, which results in very long sequences, and therefore the training process is typically time-consuming. Our method is graph-based and uses substructure-level representation for graphs, which not only captures chemical properties but also is computationally efficient.

**Substructure-level Graph Representation** Jin et al. (2018) proposed to generate molecules in the form of junction trees where each node is a ring or edge. Jin et al. (2020a) decomposed molecules into substructures by breaking all the bridge bonds. It used a complex hierarchical model for polymer generation and graph-to-graph translation. Jin et al. (2020b) proposed to extract the smallest substructure which maintains the original chemical property. The extracted substructures usually include most atoms of the original molecules, which are too coarse-grained and exert limitations on the search space. The models proposed by Jin et al. (2020a) and Jin et al. (2020b) are *not suitable* for the experiments in this paper since they either need graph-to-graph supervised data or are incompatible with continuous properties. There exists various methods to discover frequent subgraphs (Inokuchi et al., 2000; Yan & Han, 2002; Nijssen & Kok, 2004). However, they have difficulty decomposing graphs into frequent subgraphs (Jazayeri & Yang, 2021) since they mainly aim to discover frequent subgraphs as additional features for downstream network analysis. Therefore they can hardly be applied to substructure-level molecular graph representation. Different from Jin et al. (2018; 2020a) which use manual rules to extract substructures, we automatically extract common substructures which better capture the regularities in molecules for substructure-level decomposition.

## 3 APPROACH

We first give the definition of *graph pieces* and algorithms for graph piece extraction in Section 3.1. Then we describe the encoder of our graph piece variational autoencoder (GP-VAE) model in Section 3.2. Finally, we descibe the two-step decoder of GP-VAE in Section 3.3.

---

**Algorithm 1:** Graph Piece Extraction

---

**Input:** A set of graphs $\mathcal{D}$ and the desired number $N$ of graph pieces to learn.
**Result:** A set of graph pieces $\mathcal{S}$ and the counter $\mathcal{F}$ of graph pieces.

1  **begin**
2     $\mathcal{S} \leftarrow \{\text{GraphToSMILES}(\langle\{a\}, \emptyset\rangle)\};$    ▷ *Initially, $\mathcal{S}$ corresponds to all atoms $a$ that appear in $\mathcal{D}$.*
3     $N' \leftarrow \max(N, |\mathcal{S}|);$
4     **while** $|\mathcal{S}| < N'$ **do**
5        $\mathcal{F} \leftarrow \text{EmptyMap}();$                ▷ *Initialize a counter.*
6        **foreach** $\mathcal{G}$ *in* $\mathcal{D}$ **do**
7           **forall** $\langle\mathcal{P}_i, \mathcal{P}_j, \tilde{\mathcal{E}}_{ij}\rangle$ *in* $\mathcal{G}$ **do**
8              $\mathcal{P} \leftarrow \text{Merge}(\langle\mathcal{P}_i, \mathcal{P}_j, \tilde{\mathcal{E}}_{ij}\rangle);$   ▷ *Merge neighboring graph pieces into a new graph piece.*
9              $s \leftarrow \text{GraphToSMILES}(\mathcal{P});$      ▷ *Convert a graph to SMILES representation.*
10             $\mathcal{F}[s] = \mathcal{F}[s] + 1;$        ▷ *Update the counter, the default value for a new $s$ is 0.*
11          **end**
12       **end**
13       $s = \text{TopElem}(counter);$           ▷ *Find the most frequent merged graph piece.*
14       $\mathcal{P} \leftarrow \text{SMILESToGraph}(s);$        ▷ *Convert the SMILES string to graph representation.*
15       $\mathcal{S} \leftarrow \mathcal{S} \cup \{s\}; \mathcal{D}' \leftarrow \{\};$
16       **foreach** $\mathcal{G}$ *in* $\mathcal{D}$ **do**
17          $\mathcal{G}' \leftarrow \text{MergeSubGraph}(\mathcal{G}, \mathcal{P});$        ▷ *Update the graph representation if possible.*
18          $\mathcal{D}' \leftarrow \mathcal{D}' \cup \{\mathcal{G}'\};$
19       **end**
20       $\mathcal{D} \leftarrow \mathcal{D}'$
21    **end**
22 **end**

---

## 3.1 GRAPH PIECE

A molecule can be represented as a graph $\mathcal{G} = \langle \mathcal{V}, \mathcal{E} \rangle$, where $\mathcal{V}$ is a set of nodes that correspond to atoms and $\mathcal{E}$ is a set of edges that correspond to chemical bonds. Instead of using atoms, we use substructures, which we call *graph pieces*, as building blocks. We define a graph piece $\mathcal{P}$ as a subgraph $\langle \tilde{\mathcal{V}}, \tilde{\mathcal{E}} \rangle$ that appears in a graph $\mathcal{G}$, where $\tilde{\mathcal{V}} \subseteq \mathcal{V}$ and $\tilde{\mathcal{E}} \subseteq \mathcal{E}$. It should be noted that either a single atom or a whole graph is a valid graph piece. Given a set of graph pieces $\mathcal{S}$, suppose the graph $\mathcal{G}$ can be decomposed into $n$ graph pieces in $\mathcal{S}$, then $\mathcal{G}$ can be alternatively represented as $\langle \{\mathcal{P}_i\}, \{\tilde{\mathcal{E}}_{ij}\} \rangle$, where $\tilde{\mathcal{E}}_{ij}$ denotes the set of edges between two neighboring graph pieces $\mathcal{P}_i$ and $\mathcal{P}_j$. The decomposition of a graph $\mathcal{G}$ into graph pieces satisfies the following constraints: (1) the union of all atoms in the graph

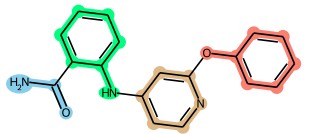

Figure 2: Four graph pieces in an example molecule. Different graph pieces are highlighted in different colors.

pieces equals to all atoms in the molecule, namely $\bigcup_i \mathcal{V}_i = \mathcal{V}$; (2) there is no intersection between any two graph pieces, namely $\forall i \neq j, \mathcal{V}_i \cap \mathcal{V}_j = \emptyset$, and $\tilde{\mathcal{E}}_i \cap \tilde{\mathcal{E}}_j = \emptyset$; (3) the union of all connections within and between graph pieces equals to all bonds in the molecule, namely $\bigcup_{i,j}(\tilde{\mathcal{E}}_i \cup \tilde{\mathcal{E}}_{ij}) = \mathcal{E}$, where $i$ range from 1 to $n$ and $j$ range from $i+1$ to $n$. Figure 2 shows an decomposed molecule.

The algorithm for extracting graph pieces from a given set of graphs $\mathcal{D}$ is given in Algorithm 1. Our algorithm draws inspiration from Byte Pair Encoding (Gage, 1994, BPE). Initially, a graph $\mathcal{G}$ in $\mathcal{D}$ is decomposed into atom-level graph pieces and the vocabulary $\mathcal{S}$ of graph pieces is composed of all unique atom-level graph pieces that appear in $\mathcal{D}$. Given the number $N$ of graph pieces to learn, at each iteration, our algorithm enumerates all neighboring graph pieces and edges that connect the two graph pieces in $\mathcal{G}$, namely $\langle\mathcal{P}_i, \mathcal{P}_j, \tilde{\mathcal{E}}_{ij}\rangle$. As $\langle\mathcal{P}_i, \mathcal{P}_j, \tilde{\mathcal{E}}_{ij}\rangle$ is also a valid subgraph, we merge it into a graph piece and count its occurrence. We find the most frequent merged graph piece $\mathcal{P}$ and add it into the vocabulary $\mathcal{S}$. After that, we also update graphs $\mathcal{G}$ in $\mathcal{D}$ that contain $\mathcal{P}$ by merging $\langle\mathcal{P}_i, \mathcal{P}_j, \tilde{\mathcal{E}}_{ij}\rangle$ into $\mathcal{P}$. The algorithm terminates when the vocabulary size exceeds the predefined number $N$. Note that we use SMILES (Weininger, 1988) to represent a graph piece in our algorithm[1], therefore we ensure the uniqueness of a graph piece. A running example of our graph piece extraction algorithm is illustrated in Figure 3. At test time, we first decompose a molecular graph into atom-level graph pieces, then apply the learned operations to merge the graph pieces into larger ones. This process ensures there is a piece-level decomposition for an arbitrary molecule. We provide the pseudo code for the piece-level decomposition in Appendix A for better understanding. We provide the complexity analysis for both algorithms in Appendix B.

---

[1]We use RDKit (www.rdkit.org) to perform the conversion between molecular graph and SMILES.

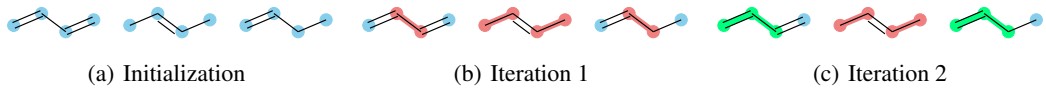

(a) Initialization                  (b) Iteration 1                  (c) Iteration 2

Figure 3: Two iterations of our graph piece extraction algorithm on {C=CC=C, CC=CC, C=CCC}. (a) The vocabulary is initialized with atoms. (b) Graph piece CC is the most frequent and added to the vocabulary. All CC are merged and highlighted in red. (c) Graph piece C=CC is the most frequent and added to the vocabulary. All C=CC are merged and highlighted in green (molecules 1 and 3). After 2 iterations the vocabulary is {C, CC, C=CC}. Note that due to the uniqueness of SMILES, graph piece C=CC will not be translated into CC=C.

## 3.2 GRAPH ENCODER

As shown in Figure 4, we use a GNN to encode a molecular graph $\mathcal{G}$ represented by graph pieces into a latent variable $z$. Each node $v$ has a feature vector $x_v$ which indicates its atomic type, the type, and generation order of the graph piece it is in. Each edge has a feature vector indicating its bond type. Modern GNNs follow a neighborhood aggregation strategy (Xu et al., 2018), which iteratively update the representations of nodes with AGGREGATE and COMBINE operations. The representation of a node $v$ in the $k$-th iteration is calculated as follows:

$$a_v^{(k)} = \text{AGGREGATE}^{(k)}(\{h_u^{(k-1)} : u \in \mathcal{N}(v)\}), \tag{1}$$

$$h_v^{(k)} = \text{COMBINE}^{(k)}(h_v^{(k-1)}, a_v^{(k)}), \tag{2}$$

where AGGREGATE and COMBINE vary in different GNNs. $\mathcal{N}(v)$ denotes neighboring nodes of $v$. We set $h_v^{(0)} = x_v$ and use GIN with edges feature (Hu et al., 2019) as the backbone GNN network, which implements AGGREGATE and COMBINE as follows:

$$a_v^{(k)} = \sum_{u \in \mathcal{N}(v)} \text{ReLU}(h_u^{(k-1)} + e_{uv}), \tag{3}$$

$$h_v^{(k)} = h_\Theta((1 + \varepsilon)h_v^{(k-1)} + a_v^{(k)}), \tag{4}$$

where $h_\Theta$ is a neural network and $\varepsilon$ is a constant. We implement $h_\Theta$ as a 2-layer multilayer perceptron (Gardner & Dorling, 1998, MLP) with ReLU activation and set $\varepsilon = 0$. Since the $k$-th representation captures $k$-hop feature of nodes (Xu et al., 2018), we obtain the final representations of nodes as $h_v = [h_v^{(1)}, \ldots, h_v^{(t)}]$ so that it contains 1-hop to $t$-hop contextual information. Then we compute the representation of the graph $\mathcal{G}$ through summation $h_\mathcal{G} = \sum_{v \in \mathcal{V}} h_v$. We use $h_\mathcal{G}$ to obtain the mean $\mu_\mathcal{G}$ and log variance $\sigma_\mathcal{G}$ of variational posterior approximation $q(z|\mathcal{G})$ through two separate linear layers and use the reparameterization trick (Kingma & Welling, 2013) to sample from the distribution in the training process.

## 3.3 TWO-STEP DECODER

With a molecule $\mathcal{G}$ represented as $\langle \{\mathcal{P}_i\}, \{\tilde{\mathcal{E}}_{ij}\} \rangle$, our model generates $\{\mathcal{P}_i\}$ and $\{\tilde{\mathcal{E}}_{ij}\}$ in two consecutive phases. The two phases move from coarse to fine granularity.

**Piece-level Sequence Generation**   Given a latent variable $z$, our model first uses an autoregressive sequence generation model $P(\mathcal{P}_i|\mathcal{P}_{<i}, z)$ to decode a sequence of graph pieces $[\mathcal{P}_1, \ldots, \mathcal{P}_n]$[2]. During training, we insert two special tokens "<bos>" and "<eos>" into the begin and the end of the graph piece sequence to indicate the begin and the end, respectively. During testing, the generation stops when a "<eos>" is generated. The sequence model can be RNNs, such as LSTM (Hochreiter & Schmidhuber, 1997) and GRU (Cho et al., 2014). In this work, we use a single layer of GRU and project the latent variable $z$ to the initial state of GRU. The training objective of this stage is to minimize the log-likelihood $\mathcal{L}_\mathcal{P}$ of the ground truth graph piece sequence:

$$\mathcal{L}_\mathcal{P} = \sum_{i=1}^{n} -\log P(\mathcal{P}_i|\mathcal{P}_{<i}, z). \tag{5}$$

_______________

[2]The ordering of graph pieces is not unique. We train with one order and leave this problem for future work.

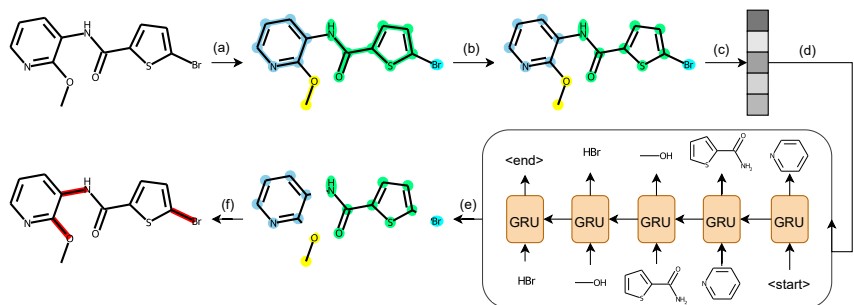

Figure 4: Overview of the *graph piece variational autoencoder*. (a) Piece-level decomposition. Atoms and bonds which belong to different graph pieces are highlighted in different colors. (b) Molecular graph. We inject piece-level information into the molecular graph through atom features. (c) Latent space encoding. We obtain the latent variable $\boldsymbol{z}$ through the graph encoder (Section 3.2). (d) Piece-level sequence generation. A sequence of graph pieces is auto-regressively decoded from the latent variable by a GRU (Section 3.3). (e) Incomplete molecular graph. The generated graph pieces form an incomplete molecular graph where inter-piece bonds are absent. (f) Bond completion. Completion of inter-piece bonds is formalized as a link prediction task for a GNN (Section 3.3). After training, we can directly sample from the latent space to generate molecules.

**Bond Completion**   We generate $\{\tilde{\mathcal{E}}_{ij}\}$ non-autoregressively. For the architecture, we use a GNN with the same structure as the graph encoder in Section 3.2 but with different parameters to obtain the representations $\boldsymbol{h}_v$ of each atom $v$ in the graph pieces. Given nodes $v$ and $u$, we predict their connections as follows:

$$P(\boldsymbol{e}_{uv}|\boldsymbol{z}) = H_\theta([\boldsymbol{h}_v; \boldsymbol{h}_u; \boldsymbol{z}]),  \tag{6}$$

where $H_\theta$ is a neural network. In this work we adopt a 3-layer MLP with ReLU activation. Apart from the types of chemical bonds, we also add a special type "<none>" to the edge vocabulary which indicates there is no connection between two nodes. During training, we predict both $P(\boldsymbol{e}_{uv})$ and $P(\boldsymbol{e}_{vu})$ to let $H_\theta$ learn the undirected nature of chemical bonds. We use negative sampling (Goldberg & Levy, 2014) to balance the ratio of "<none>" and chemical bonds. Since only about 2% pairs of nodes has inter-piece connections, negative sampling significantly improves the computational efficiency and scalability. The training objective of this stage is to minimize the log-likelihood $\mathcal{L}_{\tilde{\mathcal{E}}}$ of the ground truth inter-piece connections:

$$\mathcal{L}_{\tilde{\mathcal{E}}} = \sum_{u \in \mathcal{P}_i, v \in \mathcal{P}_j, i \neq j} -\log P(\boldsymbol{e}_{uv}|\boldsymbol{z}).  \tag{7}$$

The reconstruction loss of our GP-VAE model is:

$$\mathcal{L}_{\text{rec}} = \mathcal{L}_{\mathcal{S}} + \mathcal{L}_{\tilde{\mathcal{E}}}.  \tag{8}$$

To decode $\{\tilde{\mathcal{E}}_{ij}\}$ in inference time, the decoder first assigns all possible inter-piece connections with a bond type and a corresponding confidence level. We try to add bonds that have a confidence level higher than $\delta_{\text{th}} = 0.5$ to the molecule in order of confidence level from high to low. For each attempt, we perform a valency check and a cycle check to reject the connections that will cause violation of valency or form unstable rings which are too small or too large. Since this procedure may form unconnected graphs, we find the maximal connected component as the result of the generation. We present the pseudo code for inference in Appendix C.

We jointly train a 2-layer MLP from $\boldsymbol{z}$ to predict the scores of target properties using the MSE loss. Denote the loss of the predictor as $\mathcal{L}_{\text{prop}}$, the loss function of our GP-VAE is:

$$\mathcal{L} = \alpha \mathcal{L}_{\text{rec}} + (1 - \alpha)\mathcal{L}_{\text{prop}} + \beta D_{\text{KL}},  \tag{9}$$

where $\alpha$ and $\beta$ balance the reconstruction loss, the property loss, and the KL divergence $D_{\text{KL}}$ between the distribution of $\boldsymbol{z}$ and the prior distribution $\mathcal{N}(\boldsymbol{0}, \mathbf{I})$.

## 4 EXPERIMENTS

### 4.1 SETUP

**Evaluation Tasks**  We first report the empirical results for the distribution-learning tasks of the *GuacaMol benchmark* (Brown et al., 2019) to evaluate the ability of the model to generate realistic and diverse molecules. Then we validate our model on two goal-directed tasks. *Property Optimization* requires generating molecules with optimized properties. *Constrained Property Optimization* concentrates on improving the properties of given molecules with a restricted degree of modification.

**Dataset**  We use the ZINC250K (Irwin et al., 2012) dataset for training, which contains 250,000 drug-like molecules up to 38 atoms. For GuacaMol benchmark, we also add results on the QM9 (Blum & Reymond, 2009; Rupp et al., 2012) dataset, which has 7,165 molecules up to 23 atoms.

**Baselines**  We compare our graph piece variational autoencoder (**GP-VAE**) with the following state-of-the-art models. **JT-VAE** (Jin et al., 2018) is a variational autoencoder that represents molecules as junction trees. It performs Bayesian optimization on the latent variable for property optimization. **GCPN** (You et al., 2018) combines reinforcement learning and graph representation for goal-directed molecular graph generation. **MRNN** (Popova et al., 2019) adopts two RNNs to autoregressively generate atoms and bonds respectively. It combines policy gradient optimization to generate molecules with desired properties. **GraphAF** (Shi et al., 2020) is a flow-based autoregressive model which is first pretrained for likelihood modeling and then fine-tuned with reinforcement learning for property optimization.**GA** (Nigam et al., 2020) adopts genetic algorithms for property optimization and model the selection of the subsequent population with a neural network.

**Implementation Details**  We choose $N = 300$ for property optimization and $N = 500$ for constrained property optimization. GP-VAE is trained for 6 epochs with a batch size of 32 and a learning rate of 0.001. We set $\alpha = 0.1$ and initialize $\beta = 0$. We adopt a warm-up method that increases $\beta$ by 0.002 every 1000 steps to a maximum of 0.01. More details can be found in Appendix D.

### 4.2 RESULTS

**GuacaMol Distribution-Learning Benchmarks**  The distribution-learning benchmarks incorporate five metrics on 10,000 molecules generated by the models. *Validity* measures whether the generated molecules are chemically valid. *Uniqueness* penalizes models when they generate the same molecule multiple times. *Novelty* assesses the ability of the models to generate molecules that are not present in the training set. *KL Divergence* measures the closeness of the probability distributions of a variety of physicochemical descriptors for the training set and the generated molecules. *Fréchet ChemNet Distance (FCD)* calculates the closeness of the two sets of molecules with respect to their hidden representations in the ChemNet trained for predicting biological activities. Each metric is normalized to 0 to 1, and a higher value indicates better performance. Table 1 shows the results of distribution-learning benchmarks on QM9 and ZINC250K. Our model achieves competitive results in all five metrics, which indicates our model can generate realistic molecules and does not overfit the training set. We present some molecules sampled from the prior distribution in Appendix I.

Table 1: Results of GuacaMol distribution-learning benchmarks on QM9 and ZINC250K. KL Div refers to KL Divergence and FCD refers to Fréchet ChemNet Distance.

| Model | Validity($\uparrow$) | Uniqueness($\uparrow$) | Novelty($\uparrow$) | KL Div($\uparrow$) | FCD($\uparrow$) |
|---|---|---|---|---|---|
| *QM9* | | | | | |
| GraphAF | **1.0** | 0.500 | 0.453 | 0.761 | 0.326 |
| GA | **1.0** | 0.008 | 0.008 | 0.429 | 0.004 |
| GP-VAE(ours) | **1.0** | **0.673** | **0.523** | **0.921** | **0.659** |
| *ZINC250K* | | | | | |
| GraphAF | **1.0** | 0.288 | 0.287 | 0.508 | 0.023 |
| GA | **1.0** | 0.008 | 0.008 | 0.705 | 0.001 |
| GP-VAE(ours) | **1.0** | **0.997** | **0.997** | **0.850** | **0.318** |

**Property Optimization** This task focuses on generating molecules with optimized Penalized logP (Kusner et al., 2017) and QED (Bickerton et al., 2012). Penalized logP is logP penalized by synthesis accessibility and ring size which has an unbounded range. QED measures the drug-likeness of molecules with a range of $[0, 1]$. Both properties are calculated by empirical prediction models (Wildman & Crippen, 1999; Bickerton et al., 2012), and are widely used in previous works (Jin et al., 2018; You et al., 2018; Shi et al., 2020). We adopt the scripts of Shi et al. (2020) to calculate property scores so that the results are comparable. We directly perform gradient ascending on the latent variable (Jin et al., 2018; Luo et al., 2018). Hyperparameters can be found in Appendix D. Following previous works (Jin et al., 2018; You et al., 2018; Shi et al., 2020), we generate 10,000 optimized molecules from the latent space and report the top-3 scores found by each model. Results in Table 2 show that our model surpasses the state-of-the-art models consistently.

Table 2: Comparison of the top-3 property scores found by each model

| Method | Penalized logP ↑ | | | | QED ↑ | | | |
|---|---|---|---|---|---|---|---|---|
| | 1st | 2nd | 3rd | Validity | 1st | 2nd | 3rd | Validity |
| ZINC250K | 4.52 | 4.30 | 4.23 | 100.0% | 0.948 | 0.948 | 0.948 | 100.0% |
| JT-VAE | 5.30 | 4.93 | 4.49 | 100.0% | 0.925 | 0.911 | 0.910 | 100.0% |
| GCPN | 7.98 | 7.85 | 7.80 | 100.0% | **0.948** | 0.947 | 0.946 | 100.0% |
| MRNN | 8.63 | 6.08 | 4.73 | 100.0% | 0.844 | 0.796 | 0.736 | 100.0% |
| GraphAF | 12.23 | 11.29 | 11.05 | 100.0% | **0.948** | **0.948** | 0.947 | 100.0% |
| GA[3] | 12.25 | 12.22 | 12.20 | 100.0% | 0.946 | 0.944 | 0.932 | 100.0% |
| GP-VAE | **13.95** | **13.83** | **13.65** | 100.0% | **0.948** | **0.948** | **0.948** | 100.0% |

**Constrained Property Optimization** This task concentrates on improving the property scores of given molecules with the constraint that the similarity between the original molecule and the modified molecule is above a threshold $\delta$. Following Jin et al. (2018); You et al. (2018); Shi et al. (2020), we choose 800 molecules with lowest Penalized logP in the test set of ZINC250K for optimization and Tanimoto similarity with Morgan fingerprint (Rogers & Hahn, 2010) as the similarity metric.

Similar to the property optimization task, we perform gradient ascending on the latent variable with a max step of 80. We collect all latent variables which have better-predicted scores than the previous iteration and decode each of them 5 times, namely up to 400 molecules. Following Shi et al. (2020), we initialize the generation with sub-graphs sampled from the original molecules. Then we choose the one with the highest property score from the molecules that meet the similarity constraint. Table 3 shows our model can generate molecules with a higher Penalized logP score while satisfying the similarity constraint. Since our model uses graph pieces as building blocks, the degree of modification tends to be greater than atom-level models, leading to a lower success rate. Nevertheless, our model still manages to achieve high success rates close to atom-level models.

Table 3: Comparison of the mean and standard deviation of improvement of each model on constrained property optimization.

| Model | $\delta = 0.2$ | | $\delta = 0.4$ | | $\delta = 0.6$ | |
|---|---|---|---|---|---|---|
| | Improvement | Success | Improvement | Success | Improvement | Success |
| JT-VAE | 1.68±1.85 | 97.1% | 0.84±1.45 | 83.6% | 0.21±0.71 | 46.4% |
| GCPN | 4.12±1.19 | 100% | 2.49±1.30 | 100% | 0.79±0.63 | 100% |
| GraphAF[4] | 4.99±1.38 | 100% | 3.74±1.25 | 100% | 1.95±0.99 | 98.4% |
| GA | 3.04±1.60 | 100% | 2.34±1.34 | 100% | 1.35±1.06 | 95.9% |
| GP-VAE | **6.42±1.86** | 99.9% | **4.19±1.30** | 98.9% | **2.52±1.12** | 90.3% |

---

[3]Results are obtained by running the scripts from the original paper and use the scripts of Shi et al. (2020) to evaluate property scores.

[4]We rerun GraphAF on the same 800 molecules using its script to obtain the results since the original paper chooses a different set of 800 molecules whose results are not comparable.

### 4.3 RUNTIME COST

We train JT-VAE, GraphAF and our GP-VAE on a machine with 1 NVIDIA GeForce RTX 2080Ti GPU and 32 CPU cores and use them to generate 10,000 molecules to compare their efficiency of training and inference. As shown in Table 4, our model achieves significant improvements on the efficiency of training and inference due to graph pieces and the two-step generation. With graph pieces as building blocks, the number of steps required to generate a molecules is significantly decreased compared to the atom-level models like GraphAF. Moreover, since the two-step generation approach separates the generation of graph pieces and the connections between them into two stage and formalizes the bond completion stage as a link prediction task, it avoids the complex enumeration of possible combinations adopted by JT-VAE. Therefore, our model achieves tremendous improvement in computational efficiency over both atom-level and substructure-level baselines.

Table 4: Runtime cost for JT-VAE, GraphAF and our GP-VAE on the ZINC250K dataset. Inference time is measured with the generation of 10,000 molecules. Avg Step denotes the average number of steps each model requires to generate a molecule.

| Model | Training | Inference | Avg Step |
|---|---|---|---|
| JT-VAE | 24 hours | 20 hours | 15.50 |
| GraphAF | 7 hours | 10 hours | 56.88 |
| GP-VAE (ours) | **1.2 hours** | **0.3 hour** | **6.84** |

### 4.4 ABLATION STUDY

We conduct an ablation study to further validate the effects of graph pieces and the two-step generation approach. We first downgrade the vocabulary to contain only single atoms. Then we replace the two-step decoder with a fully auto-regressive decoder. We present the performance on property optimization and constrained property optimization after the modification in Table 5 and Table 6. The direct introduction of the two-step generation approach leads to improvement on property optimization but harms the performance on constrained property optimization. This is because separating the generation of atoms and bonds brings massive loss of bond information for the atom generation process. However, the adoption of graph pieces as building blocks alleviates this negative effect since the graph pieces themselves contain abundant bond information. Therefore, while the two-step generation approach enhances computational efficiency, the state-of-the-art performance of our model is mainly credited to the use of graph pieces.

Table 5: The top-3 property scores found by GP-VAE without certain modules.

| Method | Penalized logP | | | | QED | | | |
|---|---|---|---|---|---|---|---|---|
| | 1st | 2nd | 3rd | Validity | 1st | 2nd | 3rd | Validity |
| GP-VAE | **13.95** | **13.83** | **13.65** | 100.0% | **0.948** | **0.948** | **0.948** | 100.0% |
| - piece | 6.91 | 5.50 | 5.12 | 100.0% | 0.870 | 0.869 | 0.869 | 100.0% |
| - two-step | 3.54 | 3.54 | 3.22 | 100.0% | 0.737 | 0.734 | 0.729 | 100.0% |

Table 6: Comparison of GP-VAE without certain modules on constrained property optimization

| Model | $\delta = 0.2$ | | $\delta = 0.4$ | | $\delta = 0.6$ | |
|---|---|---|---|---|---|---|
| | Improvement | Success | Improvement | Success | Improvement | Success |
| GP-VAE | **6.42±1.86** | 99.9% | **4.19±1.30** | 98.9% | **2.52±1.12** | 90.3% |
| - piece | 2.33±1.46 | 74.8% | 2.12±1.36 | 50.9% | 1.87±1.12 | 27.0% |
| - two-step | 3.36±1.58 | 98.6% | 2.72±1.24 | 82.0% | 1.88±1.05 | 45.1% |

## 5 ANALYSIS

### 5.1 GRAPH PIECE STATISTICS

We compare the statistical characteristics of the vocabulary of JT-VAE that contains 780 substructures and the vocabulary of graph pieces with a size of 100, 300, 500, and 700. Figure 5 shows the proportion of substructures with different numbers of atoms in the vocabulary and their frequencies of occurrence in the ZINC250K dataset. The substructures in the vocabulary of JT-VAE mainly concentrate on 5 to 8 atoms with a sharp distribution. However, starting from substructures with 3 atoms, the frequency of occurrence is already close to zero. Therefore, the majority of substructures in the vocabulary of JT-VAE are actually not common substructures. On the contrary, the substructures in the vocabulary of graph pieces have a relatively smooth distribution over 4 to 10 atoms. Moreover, these substructures also have a much higher frequency of occurrence compared to those in the vocabulary of JT-VAE. We present samples of graph pieces in Appendix H.

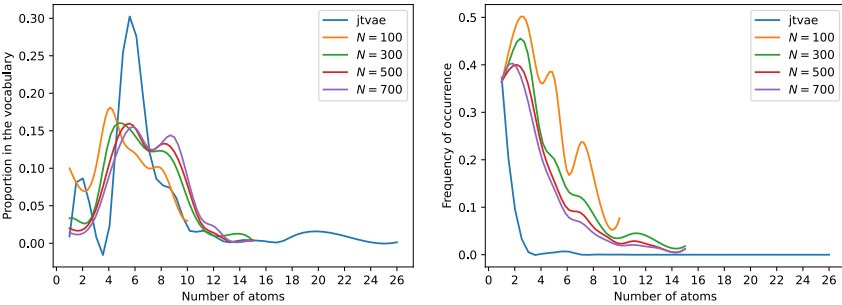

Figure 5: The left and right figures show the proportion of and frequency of occurrence of substructures with different number of atoms in the vocabulary, respectively.

### 5.2 PROPER GRANULARITY

A larger $N$ in the graph piece extraction process leads to an increase in the number of atoms in graph pieces and a decrease in their frequency of occurrence, as illustrated in Figure 6. These two factors affect model performance in opposite ways. On the one hand, the entropy of the dataset decreases with more coarse-grained decomposition (Martin & England, 2011), which benefits model learning (Bentz & Alikaniotis, 2016). On the other hand, the sparsity problem worsens as the frequency of graph pieces decreases, which hurts model learning (Allison et al., 2006). We propose a quantified method to balance entropy and sparsity. The entropy of the dataset given a set of graph pieces $\mathcal{S}$ is defined by the sum of the entropy of each graph piece normalized by the average number of atoms:

$$H_{\mathcal{S}} = -\frac{1}{n_{\mathcal{S}}} \sum_{\mathcal{P} \in \mathcal{S}} P(\mathcal{P}) \log P(\mathcal{P}), \tag{10}$$

where $P(\mathcal{P})$ is the relative frequency of graph piece $\mathcal{P}$ in the dataset and $n_{\mathcal{S}}$ is the average number of atoms of graph pieces in $\mathcal{S}$. The sparsity of $\mathcal{S}$ is defined as the reciprocal of the average frequency of graph pieces normalized by the size of the dataset:

$$S_{\mathcal{S}} = \frac{M}{f_{\mathcal{S}}}, \tag{11}$$

where $M$ is the number of molecules in the dataset and $f_{\mathcal{S}}$ is the average frequency of occurrence of graph pieces in the dataset. Then the entropy - sparsity trade-off ($T$) can be expressed as:

$$T_{\mathcal{S}} = H_{\mathcal{S}} + \gamma S_{\mathcal{S}} \tag{12}$$

where $\gamma$ balances the impacts of entropy and sparsity since the impacts vary across different tasks. We assume that $T_{\mathcal{S}}$ negatively correlates with downstream tasks. Given a task, we first sample several values of $N$ to calculate their values of $T$ and then compute the $\gamma$ that minimize the Pearson correlation coefficient between $T$ and the corresponding performance on the task. With the proper $\gamma$, Pearson correlation coefficients for the three downstream tasks in this paper are -0.987, -0.999, and

-0.707, indicating strong negative correlations. For example, Figure 6 shows the curve of entropy - sparsity trade-off with a maximum of 3,000 iteration steps for the property optimization task. From the curve, we choose $N = 300$ for the property optimization task.

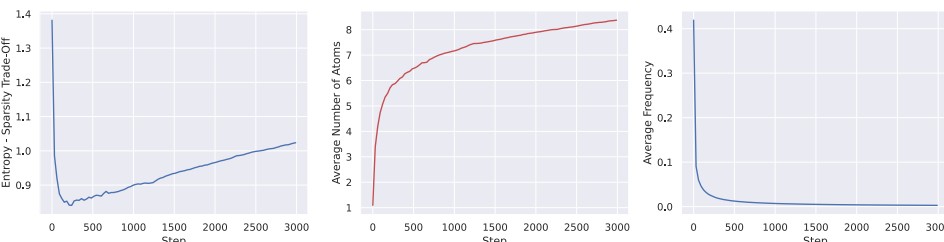

Figure 6: Entropy - Sparsity trade-off, average number of atoms in graph pieces and average frequency of occurrence of graph pieces with a maximum of 3,000 iteration steps.

### 5.3   GRAPH PIECE - PROPERTY CORRELATION

To analyze the graph piece-property correlation and whether our model can discover and utilize the correlation, we present the normalized distribution of generated graph pieces and Pearson correlation coefficient between the graph pieces and the score of Penalized logP (PlogP) in Figure 7.

The curve of the Pearson correlation coeffcient indicates that some graph pieces positively correlate with PlogP and some negatively correlate with it. Compared with the flat distribution under the non-optimization setting, the generated distribution shifts towards the graph pieces positively correlated with PlogP under the PlogP-optimization setting. The generation of graph pieces negatively correlated with PlogP is also suppressed. Therefore, correlations exist between graph pieces and PlogP, and our model can accurately discover and utilize these correlations for PlogP optimization.

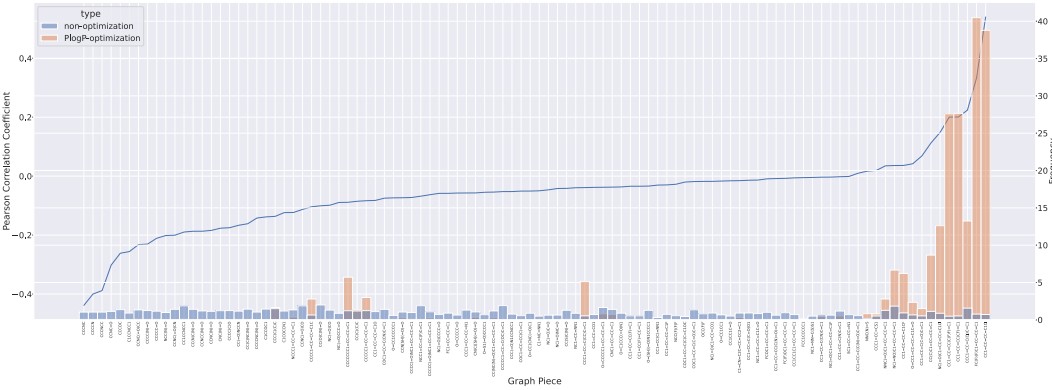

Figure 7: The distributions of generated graph pieces with and without optimization of PlogP, as well as Pearson correlation coefficient between the graph pieces and the score of PlogP. PlogP refers to Penalized logP. The distributions are normalized by the distribution of the training set, which means the frequency of occurrence of a graph piece is divided by its count of occurrence in the training set.

## 6   CONCLUSION

We propose an algorithm to automatically discover the regularity in molecules and extract them as *graph pieces*. We also propose a graph piece variational autoencoder utilizing the graph pieces and generate molecules in two phases. Our model consistently outperforms state-of-the-art models on distribution-learning, property optimization and constrained property optimization with higher computational efficiency.Our work provides insights into the selection of granularity on molecular graph generation and can inspire future search in this direction.

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

## A  PIECE-LEVEL DECOMPOSITION ALGORITHM

Algorithm 2 presents the pseudo code for the piece-level decomposition of molecules. The algorithm takes the atom-level molecular graph, the vocabulary of graph pieces and their frequencies of occurrence recorded during the graph piece extraction process as input. Then the algorithm iteratively merge the graph piece pair which has the highest recorded frequency of occurrence in the vocabulary until all graph piece pairs are not in the vocabulary.

**Algorithm 2:** Piece-Level Decomposition

**Input:** A graph $\mathcal{G}$ that decomposed into atom-level pieces, the set $\mathcal{S}$ of learned graphs pieces, and the counter $\mathcal{F}$ of learned graph pieces.

**Result:** A new representation $\mathcal{G}'$ of $\mathcal{G}$ that consists of graph pieces in $\mathcal{S}$.

```
1 begin
2 |  G' ← G;
3 |  while True do
4 |  |  freq ← −1; P ← None;
5 |  |  forall ⟨Pi, Pj, Ẽij⟩ in G' do
6 |  |  |  P' ← Merge(⟨Pi, Pj, Ẽij⟩);        ▷ Merge neighboring graph pieces into a new graph piece.
7 |  |  |  s ← GraphToSMILES(P');             ▷ Convert a graph to SMILES representation.
8 |  |  |  if s in V and F[s] > freq then
9 |  |  |  |  freq ← F[s];
10 |  |  |  |  P ← P';
11 |  |  |  end
12 |  |  end
13 |  |  if freq == −1 then
14 |  |  |  break;
15 |  |  else
16 |  |  |  G' ← MergeSubGraph(G', P);       ▷ Update the graph representation.
17 |  |  end
18 |  end
19 end
```

## B  COMPLEXITY ANALYSIS

**Graph Piece Extraction**   Since the number of graph piece pairs equals the number of inter-piece connections in the piece-level graph, the complexity is $O(NMe)$, where $N$ is the predefined size of vocabulary, $M$ denotes the number of molecules in the dataset, and $e$ denotes the maximal number of inter-piece connections in a single molecule. The number of inter-piece connections decreases rapidly in the first few iterations, therefore the time cost for each iteration decreases rapidly. It cost 6 hours to perform 500 iterations on 250,000 molecules in the ZINC250K dataset with 4 CPU cores.

**Piece-Level Decomposition**   Given an arbitrary molecule, the worst case is that each iteration adds one atom to one existing subgraph until the molecule is finally merged into a single graph piece. In this case the algorithm runs for $|\mathcal{V}|$ iterations. Therefore, the complexity is $O(|\mathcal{V}|)$ where $\mathcal{V}$ includes all the atoms in the molecule.

## C  INFERENCE ALGORITHM FOR BOND COMPLETION

Algorithm 3 shows the pseudo code of our inference algorithm. We first predict the bonds between all possible pairs of atoms in which the two atoms are in different graph pieces and sort them by the confidence level given by the model from high to low. Then for each bond with a confidence level higher then the predefined threshold $\delta_{th}$, which is 0.5 in our experiments, we add it into the molecular graph if it passes the valence check and cycle check. The valence check ensures the given bond will not violate valence rules. The cycle check ensures the given bond will not form unstable rings with nodes less than 5 or more than 6.

---

**Algorithm 3:** Inference Algorithm for Bond Completion

---

**Input:** An incomplete molecular graph $\mathcal{G}$ composed of graph pieces where inter-piece bonds are absent, the predicted bond type for all possible inter-piece connections $\mathcal{B}$ and the map to their confidence level $\mathcal{C}$, the threshold for confidence level $\delta_{th}$

**Result:** A valid molecular graph $\mathcal{G}'$

---

1 **begin**
2     $\mathcal{G}' \leftarrow \mathcal{G}$;
3     $\mathcal{B} \leftarrow \text{SortByConfidence}(\mathcal{B}, \mathcal{C})$;     ▷ *Sort the bonds by their confidence level from high to low.*
4     **forall** $b_{uv}$ *in* $\mathcal{B}$ **do**
5        **if** $\mathcal{C}[b_{uv}] < \delta_{th}$ **then**
6           continue;     ▷ *Discard edges with confidence level lower than the threshold.*
7        **end**
8        **if** valence_check($b_{uv}$) *and* cycle_check($b_{uv}$) **then**
9           $\mathcal{G}' \leftarrow \text{AddEdge}(\mathcal{G}', b_{uv})$;     ▷ *Add edges that pass valence and cycle check to $\mathcal{G}'$*
10        **end**
11     **end**
12     $\mathcal{G}' \leftarrow \text{MaxConnectedComponent}(\mathcal{G}')$;     ▷ *Find the maximal connected component in $\mathcal{G}'$*
13 **end**

---

## D  EXPERIMENT DETAILS

**Model and Training Hyperparameters**  We present the choice of model parameters in Table 7 and training parameters in Table 8. We represent an atom with three features: atom embedding, piece embedding and position embedding. Atom embedding is a trainable vector of size $e_{atom}$ for each type of atoms. Similarly, piece embedding is a trainable vector of size $e_{piece}$ for each type of pieces. Positions indicate the order of generation of pieces. We jointly train a 2-layer MLP from the latent variable to predict property scores. The training loss is represented as $\mathcal{L} = \alpha \cdot \mathcal{L}_{rec} + (1 - \alpha) \cdot \mathcal{L}_{prop} + \beta \cdot D_{KL}$ where $\alpha$ balances the reconstruction loss and prediction loss. For $\beta$, we adopt a warm-up method that increase it by $\beta_{stage}$ every fixed number of steps to a maximum of $\beta_{max}$. We found a $\beta$ higher than 0.01 often causes KL vanishing problem and greatly harm the performance. Our model and the baselines are trained on the ZINC250K dataset with the same train / valid / test split as in Kusner et al. (2017).

Table 7: Parameters in the graph piece variational autoencoder

| Model | Param | Description | Value |
|---|---|---|---|
| Common | $e_{atom}$ | Dimension of embeddings of atoms. | 50 |
| | $e_{piece}$ | Dimension of embeddings of pieces. | 100 |
| | $e_{pos}$ | Dimension of embeddings of postions. The max position is set to be 50. | 50 |
| Encoder | $d_h$ | Dimension of the node representations $\boldsymbol{h}_v$ | 300 |
| | $d_{\mathcal{G}}$ | The final representaion of graphs are projected to $d_{\mathcal{G}}$. | 400 |
| | $d_{\boldsymbol{z}}$ | Dimension of the latent variable. | 56 |
| | $t$ | Number of iterations of GIN. | 4 |
| Decoder | $d_{GRU}$ | Hidden size of GRU. | 200 |
| Predictor | $d_p$ | Dimension of the hidden layer of MLP. | 200 |

Table 8: Training hyperparameters

| Param | Description | Value |
|:---:|:---|:---:|
| $lr$ | Learning rate | 0.001 |
| $\alpha$ | Weight for balancing reconstruction loss and predictor loss | 0.1 |
| $\beta_{\text{init}}$ | Initial weight of KL Divergence | 0 |
| $\beta_{\text{max}}$ | Max weight of KL Divergence | 0.01 |
| $kl_{\text{warmup}}$ | The number of steps for one stage up in $\beta$ | 1000 |
| $\beta_{\text{stage}}$ | Increase of $\beta$ every stage | 0.002 |

**Property Optimization** We use gradient ascending to search in the continuous space of latent variable. For simplicity, we set a target score and optimize the mean square error between the score given by the predictor and the target score just as in the training process. The optimization stops if the mean square error does not drop for 3 iterations or it has been iterated to the $maxstep$. We normalize the Penalized logP in the training set to $[0, 1]$ according to the statistics of ZINC250K. By setting a target value higher than 1 the model is supposed to find molecules with better property than the molecules in the training set. To acquire the best performance, we perform a grid search with $lr \in \{0.001, 0.01, 0.1, 1, 2\}$, $maxstep \in \{20, 40, 60, 80, 100\}$ and $target \in \{1, 2, 3, 4\}$. For optimization of QED, we choose $lr = 0.01, maxstep = 100, target = 2$. For optimization of Penalized logP, we choose $lr = 0.1, maxstep = 100, target = 2$.

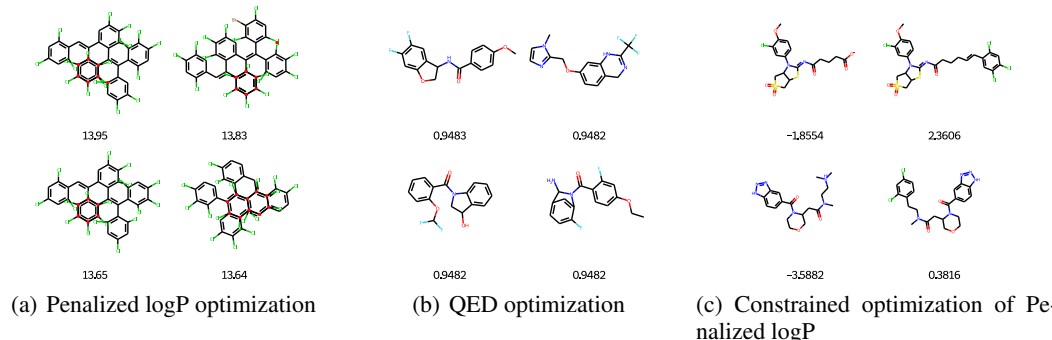

(a) Penalized logP optimization    (b) QED optimization    (c) Constrained optimization of Penalized logP

Figure 8: Samples of property optimization and constrained property optimization. In (c) the first and the second columns are the original and modified molecules labeled with their Penalized logP.

**Constrained Property Optimization** We use the same method as property optimization to optimize the latent variable. We also perform a grid search with $lr \in \{0.1, 0.01\}$ and $target \in \{2, 3\}$. We select $lr = 0.1, maxstep = 80$ and $target = 2$. For decoding, we first initialize the generation with a submol sampled from the original molecule by teacher forcing. We follow Shi et al. (2020) to first sample a BFS order of all atoms and then randomly drop out the last $m$ atoms with $m$ up to 5. We collect all latent variables which have better predicted scores than the previous iteration and decode each of them 5 times, namely up to 400 molecules. Then we choose the one with the highest property score from the molecules that meet the similarity constraint. For the baseline GA (Ahn et al., 2020), we adjust the number of iterations to 5 and the size of population to 80, namely traversing up to 400 molecules, for fair comparison.

# E    DATA EFFICIENCY

Since the graph pieces are common subgraphs in the molecular graphs, they should be relatively stable with respect to the scale of training set. To validate this assumption, we choose subsets of different ratios to the training set for training to observe the trend of the coverage of Top 100 graph pieces in the vocabularies as well as the model performance on the average score of the distribution-learning benchmarks. As illustrated in Figure 9, with a subset above 20% of the training set, the constructed vocabulary covers more than 95% of the top 100 graph pieces in the full training set, as well as the model performance on the distribution-learning benchmarks.

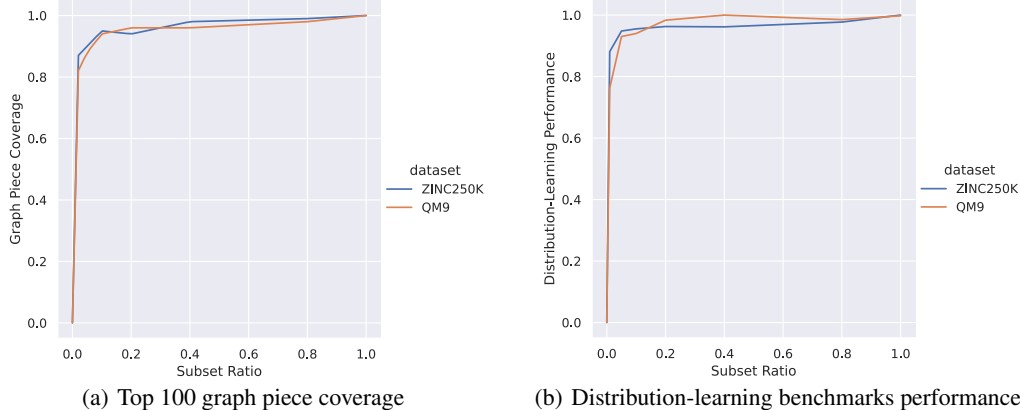

(a) Top 100 graph piece coverage    (b) Distribution-learning benchmarks performance

Figure 9: The coverage of top 100 graph pieces and the relative performance on the distribution-learning benchmarks with respect to subsets of different ratios to the full training set.

# F FUSED RINGS GENERATION

We conduct an additional experiment to validate the ability of GP-VAE to generate molecules with fused rings (cycles with shared edges), because at first thought it seems difficult for GP-VAE to handle these molecules due to the non-overlapping nature of graph pieces. We train atom-level and piece-level GP-VAEs on all 4,431 structures consisting of fused rings from ZINC250K. Then we sample 1,000 molecules from the latent space to calculate the proportion of molecules with fused rings. The results are 94.5% and 97.2% for the atom-level model and the piece-level model, respectively. The experiment demonstrates that the introduction of graph pieces as building blocks will not hinder the generation of molecules with fused rings.

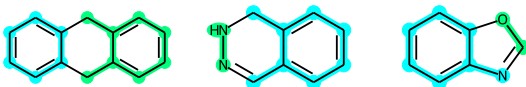

Figure 10: Decomposition of three molecules with fused rings (cycles that share edges).

# G DISCUSSION

**Universal Granularity Adaption**  The concept and extraction algorithm of *graph pieces* resemble those of subword units (Sennrich et al., 2015) in machine translation. Though subword units are designed for the out-of-vocabulary problem of machine translation, they also improve the translation quality (Sennrich et al., 2015). In this work, we demonstrate the power of graph pieces and are curious about whether there is a universal way to adapt atom-level models into piece-level counterparts to improve their generation quality. The key challenge is to find an efficient and expressive way to encode inter-piece connections into feature vectors. We leave this for future work.

**Searching in Continuous Space**  In recent years, reinforcement learning (RL) is becoming dominant in the field of optimization of molecular properties (You et al., 2018; Shi et al., 2020). These RL models usually suffer from reward sparsity when applied to multi-objective optimization (Jin et al., 2020b). However, most scenarios that incorporate molecular property optimization have multi-objective constraints (e.g.,drug discovery). In this work, we show that with graph pieces, even simple searching method like gradient ascending can surpass RL methods on single-objective optimization. It is possible that with better searching methods in continuous space our model can achieve competitive results on multi-objective optimization.

# H    GRAPH PIECE SAMPLES

We present 50 graph pieces found by our extraction algorithm in Figure 11.

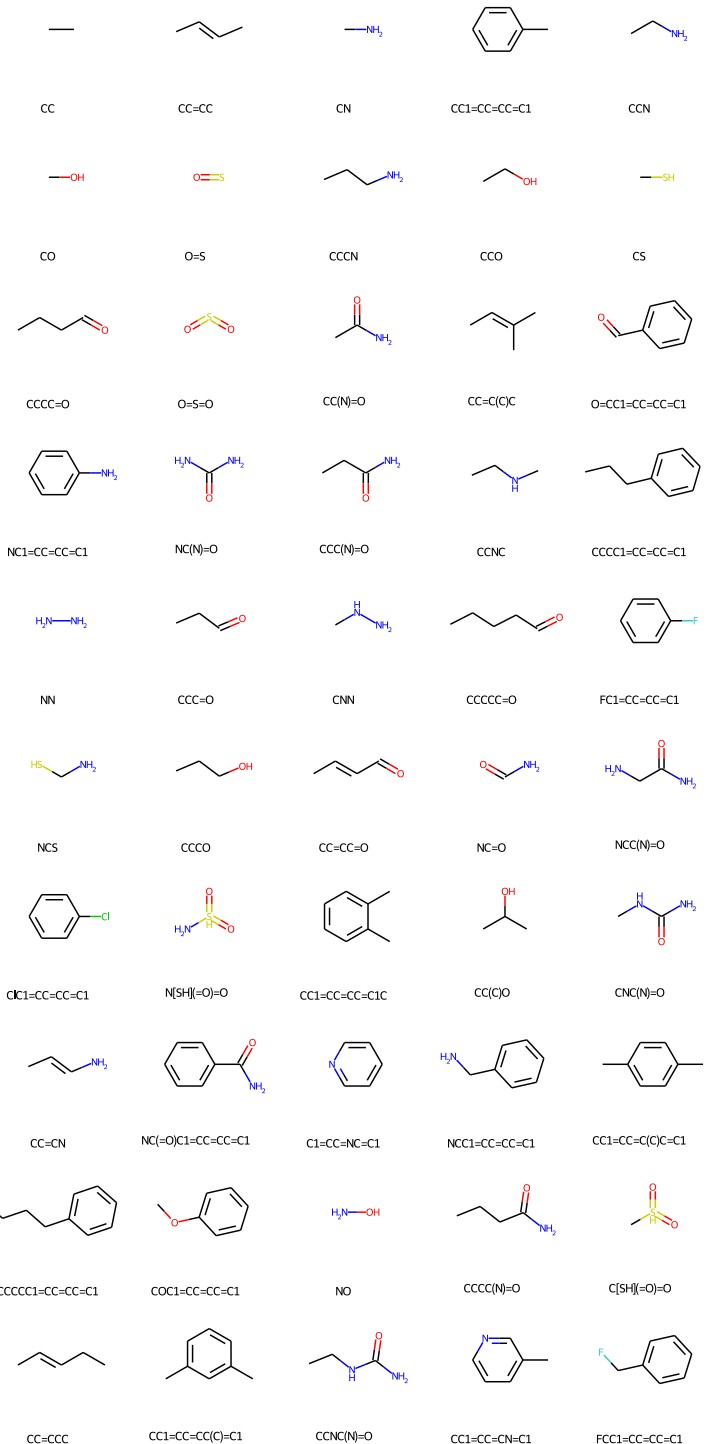

Figure 11: 50 Samples of graph pieces from the vocabulary with 100 graph pieces in total. Each graph piece is labeled with its SMILES representation.

# I MORE MOLECULE SAMPLES

We further present 50 molecules sampled from the prior distribution in Figure 12.

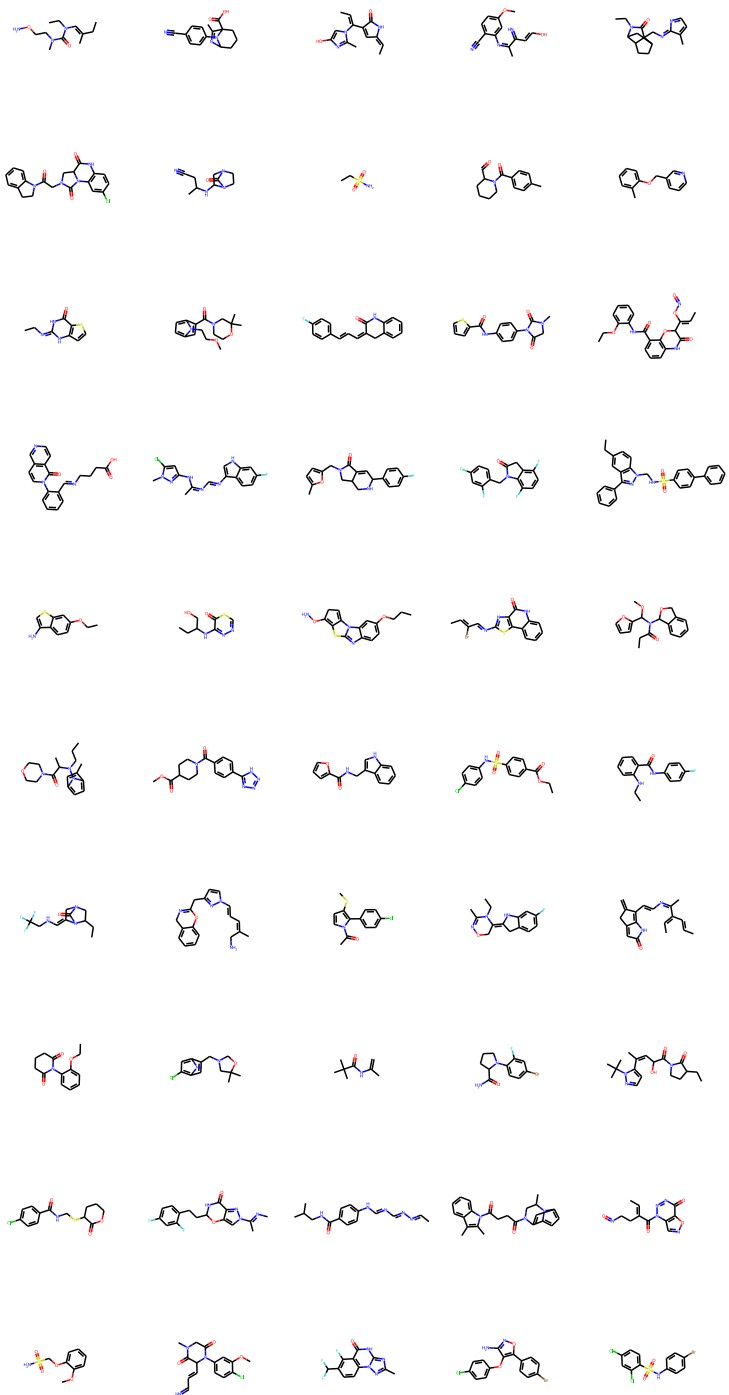

Figure 12: 50 molecules sampled from the prior distribution $\mathcal{N}(0, \boldsymbol{I})$

