# OpenReview forum: "Graph Piece: Efficiently Generating High-Quality Molecular Graphs with Substructures"
_ICLR.cc/2022/Conference — ICLR 2022 Submitted_

### Official Review · Reviewer_TrJs · 2021-10-24

**Correctness:** 3
**Technical Novelty And Significance:** 2
**Empirical Novelty And Significance:** 2
**Recommendation:** 3
**Confidence:** 4

**Main Review:**

The manuscript is unclear: how exactly is the piece level sequence generation autoregressive model connected to the bond completion? how is the fine structure (ie the nodes and edges) of each piece generated so that the bond-completion model can work only on nodes across different pieces? The summary offered is too high level and crucial information is missing (also Fig 5 is too generic).

The properties of the proposed way to generate pieces (fragments) are unclear: what is the size distribution of the fragments generated? It seems that very large fragments would be generated, but then those would be very infrequent (?), how is the tradeoff between frequency and size controlled? Why aren't there any experiments to showcase the sensitivity of the approach to the number of fragments?  What is the computational complexity of the fragment generation phase?
How often are novel fragments appearing as a function of the training set size? What are the consequences of this, with respect to the generalisation/representational capacity of the approach? i.e. the smaller the training sample the fewer the fragments (and vice versa).
Efficiency in this case seems to be in trade off with flexibility and capacity to generate novel substructures: can experiments be devised to evaluate this trade-off in a quantitative way?
Why is the vast literature of graph mining completely absent in the review section?


**Summary Of The Paper:**

The author propose a variational auto encoder for graphs based on an autoregressive model for larger fragments and a GNN for edge prediction to connect the fragments.

**Summary Of The Review:**

The justification for this work is that autoregressive models tend to ignore the existence of common substructures and hence a specific mechanism that is aware of larger fragments is needed. This fact is however never demonstrated and possibly incorrect (that is autoregressive models do not encode explicitly substructures but do this in the latent space given the compression constraints they are subject to). It would be of interest if the paper would analyse this statement empirically or theoretically, but the only results available are the usual weakly informative experiments on validity, uniqueness and novelty, and property optimization experiments (without any notion of statistical significance for the comparative results).
The presentation is overall unclear and the literature review incomplete.

---

> ### Author Response · Authors · 2021-11-23
> **Response to Reviewer TrJs**
>
> Thank you for your constructive comments! The responses to some of your concerns are listed below:
>
> Q1: how exactly is the piece level sequence generation autoregressive model connected to the bond completion? how is the fine structure (ie the nodes and edges) of each piece generated so that the bond-completion model can work only on nodes across different pieces? The summary offered is too high level and crucial information is missing (also Fig 5 is too generic).
>
> A1: We are sorry we made the description too high-level. The generation of graph pieces is similar to the generation of a sentence using RNNs, except that the token is graph pieces instead of words. Since each graph piece is a subgraph, the generated graph pieces then form an incomplete molecular graph where inter-piece bonds are absent. We use a GNN with the same structure as our encoder in Section 3.2 to predict the missing inter-piece bonds on the incomplete molecular graph, which can be formalized as a link prediction task. We have updated the description of our model structure as well as the Fig 5 (Fig 4 in the current version) for better understanding.
>
> Q2: what is the size distribution of the fragments generated? It seems that very large fragments would be generated, but then those would be very infrequent (?), how is the tradeoff between frequency and size controlled? Why aren't there any experiments to showcase the sensitivity of the approach to the number of fragments?
>
> A2: This is a good point which we missed in the original version! In Section 5.2, we have added a quantified way to analyze the trade-off between the average size of graph pieces and their frequencies when N becomes larger. We also validate the correlation between the quantified metric and the performance of downstream tasks. From the curve of this trade-off metric we are able to locate promising values of N on different downstream tasks. Also, the slope of the curve of this trade-off metric shows the sensitivity of the approach to the number of fragments to some extent.
>
> Q3: What is the computational complexity of the fragment generation phase?
>
> A3: The complexity of the graph piece extraction is O(NMe), where N denotes the size of the vocabulary, M denotes the number of molecules in the dataset and e denotes the maximum number of inter-piece connections in the molecular graphs. We also provide the complexity analysis in appendix B.
>
> Q4: How often are novel fragments appearing as a function of the training set size?
>
> A4: We have provided the trend of the coverage of top 100 graph pieces in the vocabularies when using subsets of different ratios to the training set in Appendix E. With a subset above 20% of the training set, the constructed vocabulary covers more than 95% of the top 100 graph pieces in the full training set. Therefore the common fragments are relatively stable with training sets that are not too small.
>
> Q5: What are the consequences of this, with respect to the generalisation/representational capacity of the approach? i.e. the smaller the training sample the fewer the fragments (and vice versa). Efficiency in this case seems to be in trade off with flexibility and capacity to generate novel substructures: can experiments be devised to evaluate this trade-off in a quantitative way?
>
> A5: In Appendix E, we have also provided the relative performance of our model with vocabularies extracted on subsets of different ratio to the full training set in Figure 9(b). The trend largely resembles the trend of the coverage mentioned in Q4.  From Figure 9(b) we can draw the conclusion that the performance of our model on distribution learning is also stable as long as the training set is not too small.
>
> Q6: Why is the vast literature of graph mining completely absent in the review section?
>
> A6: We are sorry we didn't explain why we didn't include methods of graph mining in the related work. The methods of frequent subgraph mining mainly aim to discover frequent subgraphs as additional features for downstream network analysis and they lack the ability to decompose an arbitrary graph into discovered frequent subgraphs. Therefore they can hardly be directly applied to substructure-level molecular graph representation. We have added the explanation and citations of frequent subgraph mining in the related work.
>
> We hope the above response could address your concerns.

---

### Official Review · Reviewer_LbDK · 2021-11-01

**Correctness:** 4
**Technical Novelty And Significance:** 3
**Empirical Novelty And Significance:** 2
**Recommendation:** 5
**Confidence:** 4

**Main Review:**

## Strength

Although each step of the proposed strategy is a straightforward application of existing methods, the overall proposed strategy is novel and interesting. In particular, it is interesting that the proposed method uses graph pieces in its training instead of directly using a given collection of graphs, which is the key idea of this paper.

Presentation of this paper is good. The paper is clearly written overall.

## Weakness

- Empirical evaluation is not thorough. Recently it is widely known that benchmarks such as the penalized logP and QED are not appropriate for evaluation and using Guacamol is highly recommended (see https://pubs.acs.org/doi/abs/10.1021/acs.jcim.8b00839).  So please evaluate the proposal on Guacamol.
- Ablation study is not convincing. An important evaluation is to compare the proposed method and that without the graph piece generation step, that is, directly using a given graph datasets in the training of GNN without decomposing them into graph pieces. This experiment tells us the effectiveness of the graph piece step.
- The proposed graph piece extraction is fully based on sequence representations of graphs, hence it does not treat all the possible subgraphs of molecular graphs. Therefore, there should be various subgraphs that cannot be extracted by the proposed method. I wonder how crucial this loss of information is. Some discussion would be desirable.

I am happy to increase my score if my concerns are addressed in the revision.


**Summary Of The Paper:**

This paper proposes a new molecular graph generation method and empirically shows its effectiveness.
The proposed method first decomposes molecular graphs into smaller parts (called graph pieces), followed by training a variational autoencoder so that it can generate the collected graph pieces. The trained model is expected to generate a variety of graphs with desirable properties.


**Summary Of The Review:**

The idea of the proposed method is technically interesting, while its empirical evaluation is not sufficient.

---

> ### Author Response · Authors · 2021-11-23
> **Response to Reviewer LbDK**
>
> Thanks for your comments and suggestions. The responses to your questions are listed below:
>
> Q1: Empirical evaluation is not thorough. Recently it is widely known that benchmarks such as the penalized logP and QED are not appropriate for evaluation and using Guacamol is highly recommended (see https://pubs.acs.org/doi/abs/10.1021/acs.jcim.8b00839). So please evaluate the proposal on Guacamol.
>
> A1: Thank you for your suggestions on the method of evaluation. The reason that we use penalized logP and QED for evaluation is that we want to follow the settings of our baselines including GraphAF, GCPN, and JT-VAE. According to your advice, we have added the results of the GuacaMol benchmark on ZINC250K and QM9 in section 4.2. We choose the distribution-learning benchmarks for evaluation due to the limitations of  time and code availability of the baselines.
>
> Q2: Ablation study is not convincing. An important evaluation is to compare the proposed method and that without the graph piece generation step, that is, directly using a given graph dataset in the training of GNN without decomposing them into graph pieces. This experiment tells us the effectiveness of the graph piece step.
>
> A2: Thank you for your advice on the design of the ablation study! We have evaluated the method without the graph piece generation step in Section 4.4.  While the two-step generation approach enhances computational efficiency, the state-of-the-art performance of our model is mainly credited to the use of graph pieces.
>
> Q3: The proposed graph piece extraction is fully based on sequence representations of graphs, hence it does not treat all the possible subgraphs of molecular graphs. Therefore, there should be various subgraphs that cannot be extracted by the proposed method. I wonder how crucial this loss of information is. Some discussion would be desirable.
>
> A3: We are sorry we didn't make it clear. Our graph piece extraction algorithm is directly implemented on the graphs. We only use the sequence representations of graphs to determine whether two graph pieces are the same since this method avoids the high complexity of the graph isomorphism problem.
>
> We hope the above response could address your concerns.

---

> > ### Comment · Reviewer_LbDK · 2021-11-24
> > **Thank you**
> >
> > Thank you for your reply. I have raised my score as some of my concerns have been successfully addressed in the revision.
> > However, I still have the following concerns:
> >
> > Although it is valuable to see evaluation on GuacaMol distribution-learning benchmarks, which I acknowledge, it is not the full set of benchmarks. Hence evaluation is still not thorough.
> >
> > Discussion of sequence representation of graphs and its relationship to subgraph mining is questionable. As other reviewers pointed out, subgraph mining is a promising and more comprehensive approach, and the statement of the authors:
> >
> > > The methods of frequent subgraph mining mainly aim to discover frequent subgraphs as additional features for downstream network analysis and they lack the ability to decompose an arbitrary graph into discovered frequent subgraphs.
> >
> > does not make sense to me as the aim of subgraph mining is actually finding subgraphs (in terms of frequency) by decomposing given graphs. More careful discussion (and potentially some comparison) would be desirable.

---

> > > ### Author Response · Authors · 2021-11-29
> > > **Thank you for your reply**
> > >
> > > Thank you for your reply! We provide further responses to your concerns as follows:
> > >
> > > 1. About other tasks in the GuacaMol benchmark:
> > >
> > > We choose the distribution-learning part of the GuacaMol benchmark is because the goal-directed part is too time-consuming as none of the baselines have provided their performance on the GuacaMol benchmark. Since most of the baselines train one model for one objective, the other 20 tasks require 20 times of training and inference of these baselines. Since most of the baselines take about 1 day to train and inference, each of the baselines takes at least 20 days to finish the whole benchmark, which is beyond the time limitation.
> > >
> > > 2. About frequent subgraph mining:
> > >
> > > Methods of frequent subgraph mining typically incorporate two processes: Candidate generation and Frequency count. Different methods adjust these two processes to improve efficiency and scalability. After counting the frequencies, these algorithms produce a set of subgraphs and tell us the percentage of graphs in the dataset containing these subgraphs. However, it is unknown how to represent these graphs with a set of frequent subgraphs not overlapping with each other. We take FSG[1] and the example of Figure 3 in our paper for illustration. The graph dataset contains 3 graphs: {C=CC=C, CC=CC, C=CCC}. FSG first finds all frequent 1-subgraphs and 2-subgraphs by enumerating all possible combinations. Therefore, the set of frequent 1-subgraphs F(1) is {C}, and the set of frequent 2-subgraphs F(2) is {CC, C=C}. To find all frequent k-subgraphs, FSG first generates candidates by joining subgraphs sharing k-2 nodes in F(k-1). Therefore, the candidates of F(3) are {C=CC}. Then FSG counts the frequency of occurrence of C=CC in the graphs and finds that 100% of the graphs contain C=CC, therefore it is a frequent 3-subgraph. After the three iterations above, FSG tells us the set of frequent subgraphs at most 3 nodes is {C, CC, C=C, C=CC}. From the process we also know their frequency of occurrence in the graphs as follows:
> > >
> > > 100% graphs contain C. 100% graphs contain C=C. 100% graphs contain CC. 100% graphs contain C=CC.
> > >
> > > However, the results cannot tell us how to represent these graphs or an arbitrary graph with a set of non-overlapping frequent subgraphs. The AGM[2] algorithm also follows this paradigm except that it only extracts induced subgraphs to reduce the complexity of the frequency counting phase. gSpan[3] searches for frequent subgraphs starting from each 1-edge toward all the children of the subgraph, which means it adds edges iteratively to the existing subgraphs to produce new candidates and then counts their frequency of occurrence. Therefore it also follows this paradigm. These methods concentrate on the discovery of frequent subgraphs with a paradigm of candidate generation followed by frequency count. Their results can tell us what are the subgraphs with a support value above the given threshold, but cannot tell us how to represent these graphs with a set of non-overlapping frequent subgraphs. In comparison, our algorithm tackles this problem by repeating the learned operations of merging the graph pieces into larger ones and then producing the decomposition of the molecular graphs.
> > >
> > > [1] M. Kuramochi and G. Karypis, “Frequent subgraph discovery.”
> > >
> > > [2] A. Inokuchi, T. Washio, and H. Motoda, “An apriori-based algorithm for mining frequent substructures from graph data.”
> > >
> > > [3] X. Yan and J. Han, “gSpan: graph-based substructure pattern mining.”

---

### Official Review · Reviewer_ogN1 · 2021-11-02

**Correctness:** 4
**Technical Novelty And Significance:** 3
**Empirical Novelty And Significance:** 3
**Recommendation:** 6
**Confidence:** 3

**Main Review:**

- strengths:


1. The idea of using common substructures to generate molecular graphs is novel and original.


2. The source code and dataset provided in supplementary material facilitate good reproducibility of this work.


3. The authors claim that the proposed method has two benefits: (1) captures correlation; (2) accelerates computation. In my opinion, it seems that the essential benefit is that, using substructure instead of atoms could reduce the search space for this combinatorial problem. Maybe the authors could theoretically analyze this.


- weaknesses:


1. I would like to know the connection and difference between graph pieces and frequent subgraph discovery. Could the authors discuss this?


2. How to obtain ground truth graph piece sequence, and why?


3. The authors state that: larger N indicates more coarse-grained decomposition. Then, how to determine proper N for different down-stream tasks?


4. The experiment is less convincing. The proposed method is only evaluated on one dataset ZINC250K. In Table 1, JT-VAE achieves perfect results, while the proposed method seems to show limited competitiveness.


5. This work misses one very related baseline [ICML 2018] [GraphRNN] Generating Realistic Graphs with Deep Auto-regressive Models, which also generates graphs in an autoregressive fashion. The authors should discuss the differences between them, and compare the performance in the experiment.


6. Figure 3 and Figure 6 are not mentioned and explained in the main text.


7. The notations are confusing. In Section 3.1, the authors state that: "$\tilde \varepsilon$ contains all the connections between the atoms in different graph pieces". In Section 3.2, the authors state that: "$\tilde \varepsilon$ contains all bonds connecting two atoms in A and B".


8. Algorithm 1 is not clear. What is the process of MolToSmiles? The authors are expected to clearly explain each line of the proposed algorithm.

**Summary Of The Paper:**

This paper proposes to use substructures instead of atoms (nodes) for generating molecular graphs.

**Summary Of The Review:**

The idea of this work is reasonable, but some technical details have not been clearly presented.

---

> ### Author Response · Authors · 2021-11-23
> **Response to Reviewer ogN1**
>
> Thank you for your comments and suggestions! The responses to your questions are listed below:
>
> Q1: I would like to know the connection and difference between graph pieces and frequent subgraph discovery. Could the authors discuss this?
>
> A1: The methods of frequent subgraph discovery (also called frequent subgraph mining) mainly aim to discover frequent subgraphs as additional features for downstream network analysis and they lack the ability to decompose an arbitrary graph into discovered frequent subgraphs. Therefore they can hardly be directly applied to substructure-level molecular graph representation. Our method not only discovers frequent subgraphs in the molecular graphs, but also ensures there is a piece-level decomposition for an arbitrary molecular graph. We have also added the explanation and citations of frequent subgraph mining in the related work.
>
> Q2: How to obtain ground truth graph piece sequence, and why?
>
> A2: We are sorry we didn't make it clear. At test time, we first decompose a molecular graph into atom-level graph pieces, then apply the learned operations to merge the graph pieces into larger ones.  This process ensures  there  is  a  piece-level  decomposition  for  an  arbitrary  molecule. We provide the pseudo code for the piece-level decomposition in Appendix A for better understanding.  After the piece-level decomposition, we can obtain the ground truth graph piece sequence from the graph.
>
> Q3: The authors state that: larger N indicates more coarse-grained decomposition. Then, how to determine proper N for different down-stream tasks?
>
> A3: In Section 5.2, we propose a quantified way to analyze the trade-off between the average size of graph pieces and their frequencies when N becomes larger. We can select proper N for different downstream tasks from the low points in the curve of this metric.
>
> Q4:  The experiment is less convincing. The proposed method is only evaluated on one dataset ZINC250K. In Table 1, JT-VAE achieves perfect results, while the proposed method seems to show limited competitiveness.
>
> A4: The reason that we only use ZINC250K dataset is that we followed the experiment setting in existing work including GraphAF, GCPN, and JTVAE. To make our experiment more convincing, we've added the results of the distribution-learning tasks of the GuacaMol benchmark on both the ZINC250K and the QM9 datasets in Section 4.2.
>
> Q5: This work misses one very related baseline [ICML 2018] [GraphRNN] Generating Realistic Graphs with Deep Auto-regressive Models, which also generates graphs in an autoregressive fashion. The authors should discuss the differences between them, and compare the performance in the experiment.
>
> A5: Thank you for your suggestion on adding GraphRNN as a baseline. Since GraphRNN is limited to generating nodes and edges without attributes, it cannot be directly used for molecular graph generation. As an alternative, we have added [MRNN] MolecularRNN: Generating realistic molecular graphs with optimized properties, which adapted GraphRNN for molecular graph generation, as a baseline for the property optimization task. Since the authors didn't publish their implementations on the github, we can only copy the results from their paper.
>
> Q6: Figure 3 and Figure 6 are not mentioned and explained in the main text.
>
> A6: Thanks for reminding us! Figure 3 (Figure 2 in the current version) is an example of decomposed molecule. Figure 6 (Figure 8 in the current version) presents the best molecules  our model found in PlogP optimization, QED optimization, and constrained optimization. We've added these in the revision.
>
> Q7: The notations are confusing. In Section 3.1, the authors state that: "$\tilde{ε}$ contains all the connections between the atoms in different graph pieces". In Section 3.2, the authors state that: "$\tilde{ε}$ contains all bonds connecting two atoms in A and B".
>
> A7: We are sorry we didn't make it clear. This notation ($\tilde{ε_{ij}}$ in the current version) indicates the set of edges between two neighboring graph pieces. We've polished the description of our methods in the revision to make it less confusing and easier to understand.
>
> Q8: Algorithm 1 is not clear. What is the process of MolToSmiles? The authors are expected to clearly explain each line of the proposed algorithm.
>
> A8: Thanks for your suggestion on improving the readability of our pseudo code. We use RDKit toolkit to do the conversion between molecular graphs and SMILES. We have rewritten the pseudo codes and added essential comments to make it easier to understand.
>
> We hope the above response could address your concerns.

---

> > ### Comment · Reviewer_ogN1 · 2021-11-23
> > **Thanks for replying**
> >
> > The authors have addressed some of my concerns, and thus I have raised my recommendation score.

---

### Official Review · Reviewer_sMpt · 2021-11-02

**Correctness:** 2
**Technical Novelty And Significance:** 2
**Empirical Novelty And Significance:** 2
**Recommendation:** 3
**Confidence:** 4

**Main Review:**

The graph piece extraction algorithm is not clear. The iteration 1 highlights the most frequent piece CC in Fig4(b). Then how C=CC in Fig4(c) is searched by merging CC and C? Line 12 of the pseudo code in appendix is not clear about how to do  “mol.merge(piece)”. Line 10 finds only the most frequent piece? Or the two most frequent pieces?  What if these two most ferquent pieces are not connectable in molecules?
And how expensive the process will be?
For molecules with rare atoms, their piece-level decomposition will be a set of single atoms?  A pseudo code algorithm will be helpful for understanding.

In Fig 5, the GNN was applied on decomposed molecule. However, section 3.3. introduces that GNN is applied on the whole graph G, like what has been done in previous GNN-based molecular graph representation.

Another concern is about the generated results. The proposed model has lower uniqueness than JT-VAE and GCPN, although very close in uniqueness metric values.

QM9 is another very popularly used dataset for molecule generation evaluation. Evaluation and comparison on this QM9 dataset can strengthen the evaluation results.

An error to correct:  We try to add bonds which has


**Summary Of The Paper:**

This is a paper working for molecule generation, especially considering the high-frequency and important subgraphs called graph pieces.

**Summary Of The Review:**

The details of the proposed model were not clearly introduced. The evaluation should be strengthened.

---

> ### Author Response · Authors · 2021-11-23
> **Response to Reviewer sMpt**
>
> Thank you very much for your constructive comments! The responses to some of your concerns are listed below:
>
> Q1: The iteration 1 highlights the most frequent piece CC in Fig4(b). Then how C=CC in Fig4(c) is searched by merging CC and C?
>
> A1: We are sorry we didn't make the demonstration clear enough. In Fig4(b) all CC are merged and therefore will be treated as single nodes in the graphs in further iterations. Thus, we can see in Fig4(b) that after all CC are merged, all piece pairs and their number of occurrences are as follows:
> (C, CC, =): 3 times, (CC, CC, =): 1 times, (C, CC, -): 1 times. Therefore C and CC connected with a double bond occurs the most frequently and will be merged in Fig4(c).
>
> Q2: Line 12 of the pseudo code in appendix is not clear about how to do “mol.merge(piece)”. Line 10 finds only the most frequent piece? Or the two most frequent pieces? What if these two most ferquent pieces are not connectable in molecules?
>
> A2: We are sorry we didn't make it clear. Line 10 only finds the most frequent piece pair and Line 12 merge the piece pair into a single piece. The definition of piece pair is a pair of neighboring graph pieces with connections between them, therefore the two pieces in a piece pair must connect with each other. We have rewritten the pseudo code and added detailed comments. Also, we move it from the appendix to Section  3.1 for better illustration.
>
> Q3: And how expensive the process will be?
>
> A3: The complexity of the process is O(NMe), where N denotes the size of the vocabulary, M denotes the number of molecules in the dataset and e denotes the maximum number of inter-piece connections in the molecular graphs. We also provide the complexity analysis in appendix B.
>
> Q4: For molecules with rare atoms, their piece-level decomposition will be a set of single atoms? A pseudo code algorithm will be helpful for understanding.
>
> A4: For molecules with rare atoms, the majority of the atoms in the molecules may also follow the regularities in other molecules, therefore their piece-level decomposition will not be a set of single atoms. For example, if we consider chlorine as a kind of rare atom, then a chlorobenzene may be decomposed into a benzene and a chlorine atom because benzene is a substructure with high frequency of occurrence in molecules. We also provide the pseudo code for the piece-level decomposition in appendix A for better understanding.
>
> Q5: In Fig 5, the GNN was applied on decomposed molecule. However, section 3.3. introduces that GNN is applied on the whole graph G, like what has been done in previous GNN-based molecular graph representation.
>
> A5: Even though we can implement an encoder only on the piece-level graph, the atom-level graph also provide essential information. Therefore we choose to fuse the piece-level information (piece division and generation order) into the atom-level molecular graph by adding the embedding and generation order of graph pieces to atom features. We found this an efficient way for the encoder to capture both piece-level and atom-level information.
>
> Q6:  Another concern is about the generated results. The proposed model has lower uniqueness than JT-VAE and GCPN, although very close in uniqueness metric values.
>
> A6: The uniqueness metric is calculated on 10,000 generated molecules. A uniqueness of 99.65% indicates only 35 of these molecules are repetitive, which means a very low possibility of repetition. We also present samples of molecules in the appendix, from which we can see there are no repetitive molecules.
>
> Q7: QM9 is another very popularly used dataset for molecule generation evaluation. Evaluation and comparison on this QM9 dataset can strengthen the evaluation results.
>
> A7: Thanks for your suggestion on more datasets. The reason that we only use ZINC250K dataset is that we followed the experiment setting in existing work including GraphAF, GCPN, and JTVAE. According to your suggestion, we've added the results of the distribution-learning tasks of the GuacaMol benchmark on the QM9 dataset in Section 4.2.
>
> Q8: An error to correct: We try to add bonds which has
>
> A8: Thank you! We have corrected it in the revision.
>
> We hope the above response could address your concerns.

---

> > ### Comment · Reviewer_sMpt · 2021-11-25
> > **REPLY**
> >
> > Thank you for the reply.
> > Why in Table 1, the baseline JT-VAE and GCPN are excluded in the revised version? They were presented in the previous submitted version.
> >
> > For Q5, the GNN was applied on the whole graph?

---

> > > ### Author Response · Authors · 2021-11-26
> > > **Response to Reviewer sMpt**
> > >
> > > The training and inference processes for these two models are too time-consuming, therefore we cannot get the results before the deadline of revision. Now we have obtained the results, which is as follows:
> > >
> > > On QM9:
> > >
> > > |Model             |Validity|Uniqueness|Novelty|KL Div|FCD|
> > >
> > > |JT-VAE           |1.0      |0.549          |0.386   |0.891  |0.588|
> > >
> > > |GCPN             |1.0      |0.533          |0.320  |0.552  |0.174|
> > >
> > > |GP-VAE(ours) |1.0      |0.673          |0.523  |0.921 |0.659|
> > >
> > > On zinc:
> > >
> > > |Model             |Validity|Uniqueness|Novelty|KL Div|FCD|
> > >
> > > |JT-VAE           |1.0      |0.988          |0.988   |0.882 |0.263|
> > >
> > > |GCPN            |1.0       |0.982          |0.982   |0.456 |0.003|
> > >
> > > |GP-VAE(ours)|1.0       |0.997          |0.997  |0.850 |0.318|
> > >
> > > Our model also achieves competitive results over these two baselines. We will add these data to the final revision if we have the chance.
> > >
> > > For Q5, the GNN is applied on the atom-level graph with piece-level information. We add the embedding and generation order of each graph piece to the features of atoms in it so that the whole graph also contains piece-level information. We choose to fuse the piece-level information into the atom-level graph due to the following two factors. First, if we apply the GNN on the piece-level graph, there exists a question: what are the types of edges in the piece-level graph? This problem is hard to solve since each edge in the piece-level graph is actually a set of bonds. If we directly assign types to the edges according to the identity of the set of bonds (begins, ends, and chemical bond types), the complexity is exponential, which is almost intractable. Second, the internal structures of graph pieces would be invisible to the encoder. Therefore we choose to fuse the piece-level information into the whole graph which is more efficient and effective.
> > >
> > > Please let us know if you have other questions. We’re happy to further answer the questions.

---

### Author Response · Authors · 2021-11-23
**Response to all the reviewers and area chair**

Firstly we would like to thank all the reviewers for your constructive reviews. We've revised the paper according to your reviews. Specifically, we have made the following changes:

1. In Section 5.2, we propose a quantified way to analyze the trade-off between the average size of graph pieces and their frequencies when N becomes larger and the granularity becomes coarser. We also validate the correlation between the quantified metric and the performance of downstream tasks. With this metric, we can select proper N for different downstream tasks.
2. In Section 5.3, we analyze the correlation between graph pieces and chemical properties, as well as whether our model can discover and utilize this correlation for property optimization.
3. We conduct additional experiments with the distribution-learning tasks of the GuacaMol benchmark on both the ZINC250K and the QM9 datasets. Results are in Section 4.2
4. We add a comparison of computational efficiency between JT-VAE, GraphAF, and our GP-VAE in Section 4.3 and analyze why our model achieves competitive results.
5. We add the ablation study on the effects of the graph piece generation step.
6. We analyze the influence of the size of the training set on the discovered graph pieces as well as the model performance in Appendix E.
7. We add the pseudo code of decomposing an arbitrary molecule into graph pieces in Appendix A. We rewrite the pseudo codes for graph piece extraction and bond completion in inference time and add comments to explain important lines. Also, we move the pseudo code of graph piece extraction from the appendix to the main text to make our algorithm easier to understand.
8. We polish the presentation of our methods and the figure of model structure to provide more detailed information.
9. We provide the complexity analysis of graph piece extraction and piece-level decomposition in Appendix B.
10. We add citations of methods of frequent graph mining and explain the connection and difference between graph pieces and frequent subgraph mining in related work.
11. We add MRNN as our baseline to the property optimization task.

---

### Decision · Program_Chairs · 2022-01-20

**Decision:**

Reject

**Comment:**

While the reviewers appreciated the new methodology and presentation of the paper the reviewers were concerned about the experimental section. Specifically they wanted to see optimization outside of penalized logP and QED, which are now viewed by the community as toy molecule optimization tasks (e.g., Penalized logP can always be improved by just adding a longer chain of carbon atoms). The authors responded that this would have taken too long to run Guacamol tasks in the rebuttal phase as all methods would need to be rerun for all tasks, but this is not true: many methods e.g., Ahn et al., 2020, already have reported these results and could be directly compared against (as this paper is near state of the art this would have been a convincing comparison). Another odd thing about the experimental setup is that the authors compare with Ahn et al., 2020 only for constrained property prediction. However Ahn et al., 2020 achieves a Penalized logP of 31.40 whereas the proposed method only achieves 13.95. It's suspicious that this result is missing in Table 2 of the current paper. If the authors are able to improve their work beyond Ahn et al., 2020 and related recent work on Guacamol and othe real-world tasks, the paper will make a much stronger submission.